# D³: DISTRIBUTIONAL DATASET DISTILLATION WITH LATENT PRIORS

## ABSTRACT

Dataset distillation, the process of condensing a dataset into a smaller synthetic version while retaining downstream predictive performance, has gained traction in diverse machine learning applications, including neural architecture search, privacy-preserving learning and continual learning. Existing methods face challenges in scaling efficiently beyond toy datasets. They also suffer from diminishing returns when increasing the distilled dataset size. We present Distributional Data Distillation (D³), a novel approach that reframes data distillation problem into a distributional one. In contrast to existing methods that distill a dataset into a finite set of real or synthetic examples, D³ produces a *probability distribution* and a decoder from which the original dataset can be approximately regenerated. We use Deep Latent Variable Models (DLVMs) to parametrize the condensed data distribution and introduce a new training objective that combines a trajectory-matching distillation loss with a distributional discrepancy term, such as Maximum Mean Discrepancy, to encourage alignment between original and distilled distributions. Experimental results across various computer vision datasets show that our method effectively distills with minimal performance degradation. Even for large high-resolution datasets like ImageNet, our method consistently outperforms sample-based distillation methods.

## 1 INTRODUCTION

The goal of data distillation, first introduced by Wang et al. (2018), is to condense a dataset into a smaller (synthetic) counterpart, such that training on this distilled dataset achieves performance comparable to training on the the original dataset. Since its inception, this problem has garnered significant attention due to its obvious implications for data storage efficiency, faster model training, and democratization of large-scale model training. It also speeds up downstream use-cases such as neural architecture search, approximate nearest neighbor search, and knowledge distillation, which require multiple iterations over the dataset (Sachdeva & McAuley (2023)). Moreover, data distillation has emerged as a promising approach for continual learning (Rosasco et al., 2021) and differential privacy (Dong et al., 2022), outperforming existing differentially-private data generators both in terms of performance and privacy, and allowing for private medical data sharing (Li et al., 2022).

Existing methods typically frame the data distillation problem as optimizing a set of samples —also called *prototypes*— of pre-defined cardinality that should summarize the dataset. These methods mostly differ in their choice of optimization objective, ranging from gradient-matching objectives (Zhao et al., 2020) to model-agnostic sample-based ones (Zhao & Bilen, 2023). While most of these methods distill into the original (e.g., pixel) space, some variants represent the prototypes in a latent space instead, and use a generator to map the latent codes back to the original data space (Deng & Russakovsky, 2022; Liu et al., 2022; Kim et al., 2022). This allows for information sharing across the prototypes, leading to better compression and higher sample quality. A similar motivation is behind methods that leverage a pre-trained generative models to aid in distillation (Cazenavette et al., 2023; Zhao & Bilen, 2022). However, these approaches have limitations. First, latent distillation methods depend heavily on the choice of latent codes and generator and do not efficiently share information across per-class generators. Generative-based methods, on the other hand, rely on pretrained generators (e.g., GANs) not specifically designed for distillation. More generally, these methods lack fine-grained control over distillation strength and often struggle to scale beyond smaller

datasets like CIFAR-10 and MNIST, experiencing diminished performance when compressing larger or higher-dimensional datasets, such as ImageNet.

In this paper, we challenge the conventional approach of distilling into a finite set of samples, instead casting the problem as a *distributional* one: finding a synthetic probability distribution which, when sampled to produce training data, yields performance comparable to training on the original dataset. To make this optimization problem tractable, we parametrize the distribution using Deep Latent Variable Models (Kingma & Welling, 2013), and design a loss function that combines a state-of-the-art gradient-matching criterion (Cazenavette et al., 2023) with a distributional loss (e.g., MMD or Wasserstein distance) — a natural choice for our distributional framework.

This novel distributional dataset distillation ($D^3$ for short) perspective is appealing because it immediately addresses many of the limitations of prior distillation methods. For example, it decouples the complexity of the distilled dataset (enforced through the class of latent distributions) from the number of samples drawn from it, which does not need to be pre-specified in our framework, and can be chosen adaptively when sampling from the distribution depending on the need (e.g., proportionally to the capacity of the model being trained). As a consequence, the memory used to store the distilled dataset (measured here in terms of the distribution parameters instead of prototype number and dimension) is more efficiently shared across samples generated from it. Additionally, this framework offers finer-grained control over the 'strength' of distillation over sample-based ones, since the number of parameters used to encode the distribution can be changed in arbitrary increments, unlike sample-based methods that can only handle increments of one sample (of dimension $d$) at a time.

Empirically, we show through various experiments that our method yields better compression rates than alternative sample- or generative-based ones. In particular, our method reaches state-of-the-art distillation methods even at a fraction of the compression rate. As a result, our distilled distribution emerges as a memory-efficient alternative to existing approaches, particularly when applied to TinyImageNet and ImageNet, where other methods struggle. Our distilled distributions exhibit robust generalization across diverse architectures not seen during training. Furthermore, by balancing computational resources and memory utilization, Finally, through a more efficient latent-decoder design, our method scales more efficiently than existing methods.

## 2    RELATED WORK

The seminal work of Wang et al. (2018), which first introduced Data Distillation, has given rise to a rapidly expanding subfield within machine learning. Here, we focus our discussion of prior work on the lines that are most closely related to ours, but note that methods with similar goals have been developed in the context of statistical sample compression (Winter, 2002; Dwivedi & Mackey, 2021) and core-set selection (Mirzasoleiman et al., 2020; Zhou et al., 2022).

**Matching Training Dynamics.**    Wang et al. (2018) originally approached the distillation problem as a bi-level optimization task, which is computationally intensive. As an alternative, Nguyen et al. (2020) leverage NTK-based algorithms to solve the inner optimization in closed form. Zhao et al. (2020) proposed gradient matching to avoid the unrolling of the inner-loop and make the distillation process more efficient. Subsequently, Zhao & Bilen (2021) propose DSA, which further improves gradient matching by performing the same image-augmentations on the target and learned data. Further improvements on single-iteration gradient matching also include Lee et al. (2022b). Matching training trajectories (MTT) was proposed by Cazenavette et al. (2023), claiming that matching long-range training dynamics provides further improvements on single-iteration gradient matching. Cui et al. (2022) proposed TESLA as a scalable alternative to the original MTT method.

**Matching Distributions.**    Distribution Matching (DM), proposed by Zhao & Bilen (2023), seeks to minimize the Maximum Mean Discrepancy (MMD) between original and distilled dataset samples. Further refinement on the method includes CAFE (Wang et al. (2022)), which explicitly aligns distributions in the feature space of a downstream task. Despite using a distributional criterion (MMD), these methods still learn a fixed number of prototypes in the original space. In contrast, we model the *prototypes themselves* as distributions, allowing for e.g., unlimited sampling from them, and leading to more diverse generation.

**Data Factorization.** In contrast to all methods listed so far, data factorization approaches distill data in a latent space (Deng & Russakovsky, 2022; Liu et al., 2022). For downstream tasks, a decoder (generator) maps the base vectors to the data space on-demand. The advantage of introducing the decoder is to allow mutual information shared by distilled examples. LinBa (Deng & Russakovsky (2022)) generates a data point using a linear combination of all the bases, while HaBa (Liu et al. (2022)) generates a data sample using a single base vector. IDC (Kim et al., 2022) stores a down-sampled version of synthetic images and conducts bi-linear upsampling in downstream training, while KFS (Lee et al. (2022a)) uses class-based latent vectors and multiple decoders to generate data.

**Generative Distillation.** This line of work learns the latent code and uses pre-trained GANs as the decoder to output training images. GLaD (Cazenavette et al. (2023)) extends MTT, and shows that optimizing in latent space promotes better cross-architecture generalizability. Building on DM (Zhao & Bilen (2023)), IT-GAN Zhao & Bilen (2022) learns a distinct latent code for every sample in the training set with the help of GAN Inversion.

## 3 DISTRIBUTIONAL DATA DISTILLATION

### 3.1 PROBLEM FORMULATION

Given a dataset $\mathcal{D} = \{\boldsymbol{x}_i, y_i\}_{i=1}^N \sim P(\mathcal{X} \times \mathcal{Y})$, the goal of dataset distillation is to find a smaller dataset such that models trained on this dataset obtain similar performance to those trained on the original one. Most existing distillation approaches can be understood as representing the distilled dataset as another finite (albeit smaller) dataset $\mathcal{S} = \{\boldsymbol{s}_i, y_i\}_{i=1}^n$, with $n \ll N$, and mostly differ in how the distilled samples $\boldsymbol{s}_i$ are represented (e.g., in the original space $\mathcal{X}$ or in a latent space $\mathcal{Z}$) and in how they are obtained (typically by gradient descent on some objective that quantifies the difference between the two datasets). A key observation from prior work is that the distilled samples need not follow the original data distribution $\mathcal{P}(\mathcal{X})$—indeed, distilling image datasets often results in synthetic samples that are not realistic images, but that nevertheless yield models that approximate those trained on the original (real) images (Wang et al., 2018).

Formally, for a class of models $f(\cdot, w) : \mathcal{X} \to \mathcal{Y}$ parameterized by their weights $w \in \mathbb{R}^D$, and a loss function $\mathcal{L}$, the objective of distillation can be phrased as finding $\mathcal{S}$ such that

$$\mathbb{E}_{\boldsymbol{x} \sim P_{\mathcal{D}}} \left[ \mathcal{L} \left( f(\boldsymbol{x}; \boldsymbol{w}^{\mathcal{S}}), y \right) \right] \simeq \mathbb{E}_{\boldsymbol{x} \sim P_{\mathcal{D}}} \left[ \mathcal{L} \left( f(\boldsymbol{x}; \boldsymbol{w}^{\mathcal{D}}), y \right) \right] \tag{1}$$

where $\boldsymbol{w}$ are the weights obtained by training (e.g., by empirical risk minimization) on either $\mathcal{D}$ or $\mathcal{S}$:

$$\boldsymbol{w}^{\mathcal{D}} = \underset{\boldsymbol{w}}{\operatorname{argmin}} \sum_{(\boldsymbol{x}_i, y_i) \in \mathcal{D}} \mathcal{L}(f(\boldsymbol{x}_i; \boldsymbol{w}), y_i), \qquad \boldsymbol{w}^{\mathcal{S}} = \underset{\boldsymbol{w}}{\operatorname{argmin}} \sum_{(\boldsymbol{s}_i, y_i) \in \mathcal{S}} \mathcal{L}(f(\boldsymbol{s}_i; \boldsymbol{w}), y_i) \tag{2}$$

As in prior distillation work, here we focus on classification with deep neural networks, so we assume $\mathcal{L}$ to be the cross-entropy loss and take $f(\cdot; \boldsymbol{w})$ to be a neural network with weights $\boldsymbol{w}$.

Although casting the problem using the finite dataset $\mathcal{S}$ follows naturally from the empirical objectives (2), it is certainly not the only way to conceptualize distillation. For example, considering the population counterparts of (2), i.e.,

$$\underset{\boldsymbol{w}}{\operatorname{argmin}} \underset{(\boldsymbol{x}_i, y_i) \sim P}{\mathbb{E}} \mathcal{L}(f(\boldsymbol{x}_i; \boldsymbol{w}), y_i), \qquad \underset{\boldsymbol{w}}{\operatorname{argmin}} \underset{(\boldsymbol{x}_i, y_i) \sim Q}{\mathbb{E}} \mathcal{L}(f(\boldsymbol{x}_i; \boldsymbol{w}), y_i), , \tag{3}$$

we can instead formulate the problem as finding a synthetic distribution $Q$ which, as before, leads to comparable predictive performance for $f$. To make the problem tractable, let us consider a family of distributions $Q_\theta$ parameterized by $\theta \in \Theta \subseteq \mathbb{R}^D$. This in turn allows us to formulate the problem as an optimization over the finite-dimensional $\Theta$ rather than the infinite-dimensional space of distributions:

$$\underset{\theta}{\min} \quad \mathcal{D}(\boldsymbol{w}^{\mathcal{S}}(\theta), \boldsymbol{w}^Q; P) \qquad s.t. \quad \boldsymbol{w}^{\mathcal{S}}(\theta) = \underset{\boldsymbol{w}}{\operatorname{argmin}} \underset{(\boldsymbol{x}_i, y_i) \sim Q_\theta}{\tilde{\mathbb{E}}} \mathcal{L}(f(\boldsymbol{x}_i; \boldsymbol{w}), y_i), \tag{4}$$

where $\mathcal{D}$ is some loss function between model weights and $\tilde{\mathbb{E}}$ denotes an empirical expectation, which in contrast to (2), does not a-priory specify which and how many samples it is computed over. Note that here we make explicit the dependence of the post-distillation model parameters $\boldsymbol{w}^{\mathcal{S}}(\theta)$ on the distributional parameters $\theta$ for clarity, but we will henceforth simply write $\boldsymbol{w}^{\mathcal{S}}$ for brevity.

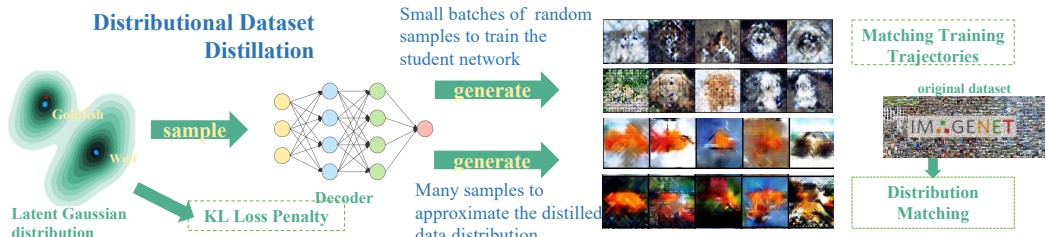

**Figure 1:** Our method distills a dataset into a synthetic latent distribution that, when sampled from to train models, yields comparable predictive performance to the original dataset.

At this point, it is important to note a subtle but crucial difference between the traditional (sample-based) and this (distributional-based) approaches to dataset distillation. Recall that one of the main goals of distillation is to reduce the size of the original dataset. When distilling into another dataset, the 'compression' aspect is clear: $n = |\mathcal{S}|$ should be taken to be smaller than $N = |\mathcal{D}|$. When instead distilling into a distribution —as we propose here— compression is far more subtle to define since $\mathcal{D}$ and $Q$ are no longer directly comparable. We argue that in this case 'distillation' is satisfied if (i) the storage footprint (measured in terms of cardinality, or more precisely, a unit of memory size, e.g., bytes) of $\theta$ is smaller than that of $\mathcal{D}$ and (ii) the effective number of samples from $Q$ on which a model needs to be trained is comparable, or lower, than that of training on $\mathcal{D}$. In particular, we seek to avoid the two trivial corner-cases $Q_\theta = \frac{1}{N} \sum_{i=1}^{N} \delta_{\boldsymbol{x}_i}$, i.e., the uniform empirical measure associated with $\mathcal{D}$, and $Q_\theta \approx P$, i.e., learning the full distribution $P$ — a much harder problem to solve. Our solution to this challenge is to enforce compression by restricting the complexity of $Q_\theta$ through a specific choice of parametric representation that we discuss in the next section. This choice of representation has the dual effect of addressing storage (by giving us fine-grained control over the number of parameters required to represent the distribution) and training time (by reducing the sample complexity of the distilled estimation problem). To quantify the former, we discuss the notion of compression rate for this distributional setting in Section 3.4.

## 3.2 PARAMETRIZING THE DISTILLED DISTRIBUTION

Our parametrization of the distilled distribution takes inspiration from Deep Latent Variable Models (DLVMs, Kingma & Welling (2019)), a flexible family of statistical models that combine the foundational principles of variational inference with the approximation power of deep neural networks. DVLMs have the advantage of encoding information into a latent (typically lower-dimensional) space and provide a method to map from it to data space, enabling flexible generation. Additionally, other distillation methods based on latent factorizations often achieve better distillation performance than explicit sample-based ones Zhao & Bilen (2022); Cazenavette et al. (2023); Lee et al. (2022a). However, these methods still pose the distillation problem in the usual sample-based form, in terms of finding a set of pre-determined (and, in particular, finite) size of (latent) prototypes. This implies, in particular, that they are restricted to a one-to-one mapping between latent and explicit prototypes.

Here, we leverage the benefits of latent variable models but adapt them to the distributional formulation of the problem introduced in the previous section. Concretely, we represent the distilled dataset directly as a probability distribution, parametrized as a DVLM. Formally, we model the distilled distribution in a variational form as

$$Q_{\mathcal{S}}^\theta(\boldsymbol{x}) = \int Q_{\mathcal{S}}^\theta(\boldsymbol{x}|\boldsymbol{z}) p(\boldsymbol{z}) \, \mathrm{d}\boldsymbol{z},$$

where $Q_{\mathcal{S}}^\theta(\boldsymbol{x}|\boldsymbol{z})$ is parametrized by a neural network and maps the prior $z$ to the distilled samples $\boldsymbol{x}$.

We may assume that each class shares one prior distribution, $p(\boldsymbol{z}|c) \sim \mathcal{N}(\mu_c, \Sigma_c)$, with the parameter pair $(\mu_c, \Sigma_c)$ learned during the data distillation process. Therefore, data points for each class follow a multivariate Gaussian distribution. Additionally, we also experiment multi-modal distribution by allowing multiple Gaussian priors for each class. We refer to the number of Gaussian priors for each class as Priors Per Class (PPC). Under the multi-modal set up, for each class $c$, the set of all possible prior distributions $\{\mathcal{N}(\mu_c^i, \Sigma_c^i)\}_{i=1}^{\mathrm{PPC}}$ follows a uniform distribution.

### 3.3 Learning the Distilled Distribution

Building on the foundations of existing data distillation techniques, we introduce a learning objective comprised of two distinct terms. The first term is derived from Matching Training Trajectories (MTT) proposed by Cazenavette et al. (2022). The second term in our objective is based on MMD, using both to match distribution between the true dataset and our learned dataset distribution and to encourage diversity within the distill distribution.

**MTT Loss**   Expert trajectories are training trajectories generated from training neural networks on the full training set. At each distillation step, we initialize a student network that has the same architecture as the experts. The student network's initialization weight $\boldsymbol{w}^Q$ is sampled from the experts training trajectory by randomly selecting an expert and a random iteration $t$, such that $\boldsymbol{w}_t^Q = \boldsymbol{w}_t^{\mathcal{P}}$. We perform $N$ gradient updates on the student network using data drawn from the distilled distribution:

$$\text{for } n = 0....N - 1 : \boldsymbol{w}_{t+n+1}^Q = \boldsymbol{w}_{t+n}^Q - \alpha \nabla \mathcal{L}(Q; \boldsymbol{w}_{t+n}^Q), Q \sim Q_{\mathcal{S}}^\theta$$

We then collect expert parameters from $M$ training updates after iteration $t$, which denote as $\boldsymbol{w}_{t+M}^{\mathcal{D}}$. The distance between the updated student parameters and the updated expert parameters is quantified using normalized squared error:

$$D_{\text{MTT}} = \frac{\|\boldsymbol{w}_{t+N}^Q - \boldsymbol{w}_{t+M}^{\mathcal{D}}\|_2^2}{\|\boldsymbol{w}_t^{\mathcal{P}} - \boldsymbol{w}_{t+M}^{\mathcal{D}}\|_2^2}$$

An additional KL-Divergence term is incorporated to align the latent distribution with the standard Gaussian distribution. The final MTT loss consists of the MTT distance and the KL penalty, weighted by hyperparameter $\gamma_1$.

$$\mathcal{L}_{\text{MTT}} = \frac{\|\boldsymbol{w}_{t+N}^Q - \boldsymbol{w}_{t+M}^{\mathcal{D}}\|_2^2}{\|\boldsymbol{w}_t^{\mathcal{P}} - \boldsymbol{w}_{t+M}^{\mathcal{D}}\|_2^2} + \gamma_1 \sum_{c=1}^C \text{KL}\left(\mathcal{N}\left(\mu_c, \Sigma_c\right) \| \mathcal{N}\left(0, 1\right)\right)$$

**MMD**   Based on the formulation used in DM (Zhao & Bilen, 2023), we additonally use a set of Reproducing Hilbert Kernels (RHKS) for the MMD computation to fully leverage the power of MMD. Since we only have access to the distilled distribution $Q_{\mathcal{S}}^\theta$ but not the training data distribution $P$, we use the empirical MMD measure: In general, given random variable $X = \{x_1, ..., x_n\} \sim \mathbb{P}$ and $Y = \{y_1, ..., y_m\} \sim \mathbb{Q}$, the unbiased estimator of the MMD measure is (Li et al. (2017)):

$$\widehat{\text{MMD}}^2(X, Y) = \frac{1}{\binom{n}{2}} \sum_{i \neq j}^n k(x_i, x_j) - \frac{1}{mn} \sum_{i=1}^n \sum_{j=1}^m (x_i, y_j) + \frac{1}{\binom{m}{2}} \sum_{i \neq j}^m k(y_i, y_j) \tag{5}$$

We also map the pixel space to latent feature space. For model simplicity and training efficiency purposes, we recycle the experts used in MTT to generate feature mappings, and denote them as $\psi(\cdot)$. Inspired by MMD GANs (see Li et al. (2017); Bińkowski et al. (2018)), we use a mixture of Radial Basis Function (RBF) kernels $k(x, x') = \sum_{q=1}^K k_{\sigma_q}(x, x')$, where $k_{\sigma_q}$ represents a Gaussian kernel with bandwidth $\sigma_q$. We choose a mixture of $K = 5$ kernels with bandwidths $\{1, 2, 4, 8, 16\}$.

To encourage distribution matching with the original dataset, we penalize large MMD:

$$\mathcal{L}_{\text{MMD}} = \sum_{c=1}^C \widehat{\text{MMD}}^2(\psi(\mathcal{D}_c), \psi(\mathcal{S}_c)), \tag{6}$$

where the $\widehat{\text{MMD}}^2$ computation is defined in Eqn. 5. $\mathcal{D}_c$ and $\mathcal{S}_c$ simply refers to the subset of the training data or distilled data with class label $c$.

**Diversity**   For multi-modal cases, where we choose to use multiple prior distributions to represent each class, we want the distributions from the same class to be orthogonal from each other so that they each represent a set of distinct information for each class. Therefore, we impose a diversity loss to encourage large pairwise MMD measure among the prior distributions from the same class.

Our DM loss encompasses the MMD loss and the Diveristy loss:

$$\mathcal{L}_{\text{MMD}} = \gamma_2 \sum_{c=1}^{C} \widehat{\text{MMD}}^2(\psi(\mathcal{D}_c), \psi(\mathcal{S}_c)) - \frac{\gamma_3}{\text{PPC}} \sum_{i \neq j}^{\text{PPC}} \widehat{\text{MMD}}^2(\psi(\mathcal{S}_c^i)), \psi(\mathcal{S}_c^j)),$$

where $\mathcal{S}_c^i$ refers to the distilled examples with class label $c$ and have prior distribution $\mathcal{N}\left(\mu_c^i, \Sigma_c^i\right)$. We use relative weights $\gamma_2$ and $\gamma_3$ to balance the scale of each loss term. Refer to Algorithm 1 in Appendx A for an overview of our main algorithm.

### 3.4 Measuring the Compression Rate of Distributional Distillation

Compression rate is defined as the ratio between the storage footprint of the distilled dataset (or distribution) and of the entire training set. For traditional data distillation methods that directly output samples, the compression rate is well-defined: Comp. Rate = (Distilled sample size)/(Training set size). For factorization methods, the current convention is to only accounts for the memory it takes to store the latent codes and the decoder. Since our framework outputs distributions, we adopt the convention from factorization methods to measure our compression rate. However, compression rate does not capture all aspects of data distillation. It does not address the cost of compute during inference to generate samples. Furthermore, it also does not capture the total number of samples factorization methods and our methods need to generate to achieve good performance on downstream tasks.

### 3.5 Memory Constraints

The data distillation process itself can be memory costly, especially on large datasets. There are three factors during training that creates memory bottlenecks: the number of prototypes learned, the number of steps student networks takes in the MTT loss, and the number of samples used to compute MMD loss. To mitigate the memory cost of entire pipeline, we divide up the two training losses such that the backward propagation step is taken on the distill distribution right after each loss term. Such a separation allows for the compute graph to be cleared to free up more memory for the next training step. To compute the MMD loss, one would ideally use the entire training set to reduce variance. To mitigate memory cost, we perform MMD computations on small batches. For all our experiments, we used at most four NVIDIA A100-SXM4-40GB GPUS to perform distillation.

## 4 Experiments

Our experimental evaluation considers three distinct datasets: CIFAR-10, TinyImageNet, and ImageNet. The objective of our experiments is to conduct comparative evaluations of our methodologies vis-à-vis relevant optimization methods (MTT and DM), factorization methods (HaBa, LinBa, and KFS), and state-of-the-art generative methods (IT-GAN and GLaD). For each prior distribution, we use a 64-dimensional Gaussian. We adopt our decoder from convolutional VAE proposed by Li et al. (2017). For RGB images with size from $32 \times 32$ to $128 \times 128$, the decoder has 750K parameters.

In pursuit of impartial comparisons with existing data distillation methodologies, we align all our design choices with existing work. For all our datasets, we use ConvNet architecture for our distillation tasks. We provide a detailed description of the decoder and size breakdown in Appendix B and further details on the rest of the experimental set-up in Appendix C.

### 4.1 From small datasets to large datasets

**Comparison with State of the Arts** The comparison results against MTT and DM are shown in Table 2. We observe that for CIFAR-10, our method achieves comparable performance as MTT's 50 Images Per Class (IPC) case with a compression rate $2\times$ more efficient. For TinyImageNet, the best distillation quality our method provides ($\sim 24\%$) is comparable to MTT with 10 IPC but we improve the compression rate $4\times$. However, our method has yet to scale to MTT's performance at 50 IPC. The comparison results against HaBa, LinBa and KFS are shown in Table 2. In the case of TinyImageNet, our method achieves a compression rate nearly five times superior to LinBa. For ImageNet subsets, we our method achieves a superior distillation quality at the cost that is five times smaller than HaBa.

**Table 1:** Test Accuracy (%) of a model trained on datasets generated with various distillation methods using the same architecture (ConvNet). PPC= Images (DM, MTT) or Priors (Ours) Per Class.

| Dataset | Distillation Method | | | | | | Full |
|---|---|---|---|---|---|---|---|
| | MTT | | DM | | Ours | | |
| PPC* | Comp.Rate (%) | Test Acc | Comp. Rate (%) | Test Acc | Comp.Rate (%) | Test Acc | Test Acc |
| **CIFAR10** | | | | | | | 84.8 (0.1) |
| 1 | 0.02 | 46.3 (0.8) | 0.02 | 26.0 (0.8) | 0.5 | 60.61 (0.14) | |
| 2 | — | — | — | — | 0.5 | 64.98 (0.14) | |
| 3 | — | — | — | — | 0.51 | 66.64 (0.26) | |
| 10 | 0.2 | 65.3 (0.7) | 0.2 | 48.9 (0.6) | 0.51 | **71.79 (0.19)** | |
| 50 | 1 | **71.6 (0.2)** | 1 | 65.2 (0.4) | — | — | |
| **TinyImageNet** | | | | | | | 37.6 (0.4) |
| 1 | 0.2 | 8.8 (0.3) | 0.2 | 3.9 (0.2) | 0.06 | 19.81 (0.34) | |
| 2 | — | — | — | — | 0.07 | 24.61 (0.26) | |
| 10 | 2.0 | 23.2 (0.2) | 2.0 | 12.9 (0.4) | — | — | |
| 50 | 10 | **28.0 (0.3)** | 10 | 24.1 (0.3) | — | — | |
| **ImageNette** | | | | | | | 87.4 (0.1) |
| 1 | 0.08 | 47.7 (0.9) | — | — | 0.12 | 67.44 (0.81) | |
| 2 | — | — | — | — | 0.12 | **71.04 (0.71)** | |
| 10 | 0.78 | 63.0 (1.3) | — | — | — | — | |
| **ImageWoof** | | | | | | | 67.0 (1.3) |
| 1 | 0.08 | 28.6 (0.8) | — | — | 0.13 | 36.52 (0.77) | |
| 2 | — | — | — | — | 0.13 | **41.60 (1.15)** | |
| 10 | 0.8 | 35.8 (1.8) | — | — | — | — | |

**Qualitative Analysis.** We present samples from our distilled distribution in Figure 2. By feeding the distilled mean $\mu_c$ directly to the decoder, we can visualize "prototypical" examples for each class. The two typical mushroom samples are of very different style and their variations further modify on the angle, shape for example. In Figue 2, we also show how "typical" samples evolves as we increase the number of priors.

**Table 2:** Downstream test Accuracy (%) of various latent-encoding methods on ImageNet variants.

| Dataset | CR (%) | Method | | | |
|---|---|---|---|---|---|
| | | LinBa | HaBa | KFS | Ours |
| **TinyImageNet** | 0.06 | — | — | — | 19.81 (0.34) |
| | 0.07 | — | — | — | 24.61 (0.26) |
| | 0.2 | 16.0 (0.7) | | 22.7 (0.2) | |
| | 2.0 | — | | **27.8 (0.2)** | |
| **ImageNette** | **0.12** | — | — | — | **71.04 (0.71)** |
| | 0.16 | — | 51.92 (1.65) | — | |
| | 0.78 | — | 64.72 (1.60) | — | |
| **ImageWoof** | **0.13** | — | — | — | **41.60 (1.15)** |
| | 0.16 | — | 32.40 (0.67) | — | |
| | 0.78 | — | 38.60 (1.26) | — | |

**Scaling to larger sizes.** Table 3 compares our method's scaling properties on CIFAR-10 with factorization and generative-based methods. The compression rate for our methods corresponds to $\{1, 2, 3, 10\}$ PPC. As before, our approach exhibits diminishing returns when adding more prior distributions per class, but the marginal cost in doing so is minimal compared to the cost (paid once) of having a shared decoder. As a result, compared to factorization methods (Haba. LinBa and KFS), our method scales better. KFS requires mapping every training sample to a latent code during distillation, but our method provides the flexibility in choosing the number of priors a priori based on desired distillation quality. We also train a decoder tailored to our prior distribution. As a result, our method provides finer-grained control on the compression rate than IT-GAN.

## 4.2 FROM SEEN ARCHITECTURE TO UNSEEN ARCHITECTURES

A data distillation method can only be practical if it demonstrates the capability to generalize effectively across architectures. We use distilled distribution trained on ConvNet to train models with unseen architectures, including ResNet18, VGG, AlexNet, and Vision Transformer. Adding to previously mentioned baselines, we also add GLaD Cazenavette et al. (2023), which is an extension to MTT specifically to enhance the cross-architecture performance.

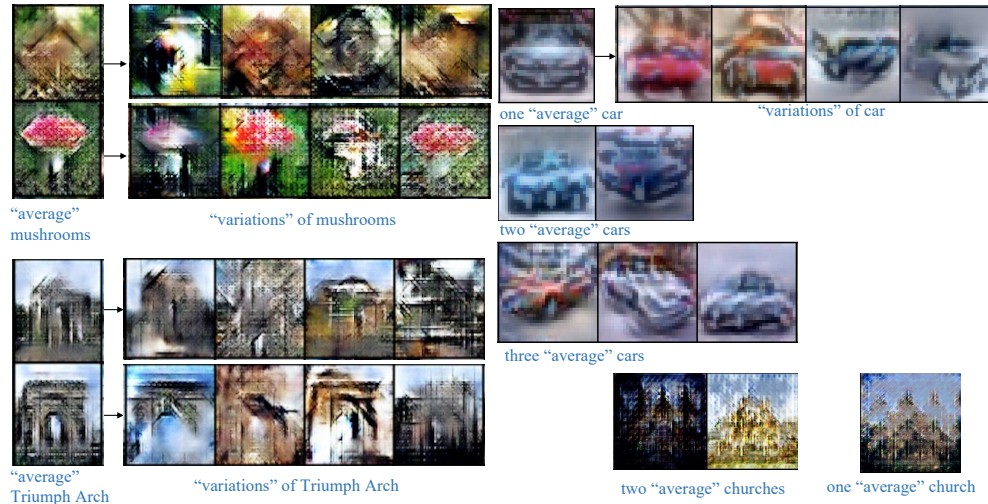

**Figure 2: Upper left**: distilled TinyImageNet mushrooms under 2 PPC setting. Average image generated by passing the mean of the prior $\mu_c$ to the decoder. Variations images generated by sampling $z \sim \mathcal{N}(\mu_c, \Sigma_c)$ and passing to the decoder. **Lower left**: TinyImageNet Triumph Arch. **Upper Right**: three sets of distilled average CIFAR-10 cars from $\{1, 2, 3\}$ PPC settings. **Lower Right**: two sets of distilled ImageNette church from $\{1, 2\}$ PPC settings.

Table 4 presents our results on CIFAR-10 in comparison to MTT and DM. When comparing our method to MTT, we observe a clear improvement in cross-architecture generalization. Compared to DM, we observe similar cross-archiecture performance but we achieve it at a compression rate twice as efficient. In Appendix D.1, we show that the distribution matching loss is crucial in improving the cross architecture performance. In Table 5, we perform

**Table 3:** Downstream test accuracy (%) comparison to factorization and generative methods on CIFAR-10. Our method scales better than Haba, LinBa and KFS, and provides finer-grained control than IT-GAN.

| Method | Comp. Rate (%) | | |
|---|---|---|---|
| | 0.02 | 0.2 | 1 |
| HaBa | 48.3 (0.8) | 69.9 (0.4) | 74.0 (0.2) |
| LinBa | 66.4 (0.4) | 71.2 (0.4) | 73.6 (0.5) |
| KFS | 59.8 (0.5) | 72.0 (0.3) | 75.0 (0.2) |

| Method | Comp. Rate(%) | | | |
|---|---|---|---|---|
| | 0.504 | 0.505 | 0.506 | 0.511 |
| Ours | 60.61 (0.1) | 64.98 (0.1) | 66.64 (0.2) | 71.79 (0.2) |

| Method | Comp. Rate(%) | Test. Acc. |
|---|---|---|
| IT-GAN | 25 | 82.8 (0.3) |

comparisons on ImageNet subsets with available results from GLaD (only 1 PPC available).

**Table 4:** CIFAR-10 cross-architecture generalization (test accuracy %) comparison to MTT and DM.

| | Distillation Level | | Evaluation Model | | | |
|---|---|---|---|---|---|---|
| Method | PPC | Comp. Rate(%) | ConvNet | ResNet18 | VGG11 | AlexNet |
| MTT | 10 | 0.2 | 65.3 (0.7) | 46.4 (0.6) | 50.3 (0.8) | 34.2 (2.6) |
| DM | 50 | 1 | 65.2 (0.4) | 57.0 (0.9) | 59.9 (0.8) | 61.3 (0.6) |
| | 1 | 0.5 | 60.61 (0.14) | 54.99 (0.72) | 53.83 (0.55) | 46.88 (0.7) |
| | 2 | 0.5 | 64.98 (0.14) | 61.07 (0.30) | 59.50 (0.53) | 51.80 (1.80) |
| Ours | 3 | 0.51 | **66.64 (0.26)** | **61.57 (0.48)** | **59.70 (0.48)** | **54.56 (0.74)** |

**Table 5:** ImageNet Subset cross-architecture performance (Test Accuracy %) comparison to MTT and GLaD. Unseen archiecture results from averaging ResNet18, VGG11, AlexNet, Vision Transformer.

| Method | Distillation Level | | ImageNette | | ImageWoof | |
|---|---|---|---|---|---|---|
| | PPC | Comp.Rate(%) | ConvNet | Unseen | ConvNet | Unseen |
| MTT | 1 | 0.08 | 47.9 (0.9) | 24.1 (1.8) | 28.6 (0.8) | 16.0 (1.2) |
| GLaD MTT | 1 | 0.08 | 38.7 (1.6) | 30.4 (1.5) | 23.4 (1.1) | 17.1 (1.1) |
| GLaD DC | 1 | 0.08 | 35.4 (1.2) | 31.0 (1.6) | 22.3 (1.1) | 17.8 (1.1) |
| GLaD DM | 1 | 0.08 | 32.3 (1.7) | 21.9 (1.1) | 21.1 (1.5) | 15.2 (0.9) |
| Ours | 1 | 0.12 | **67.44** (0.81) | **48.95** (1.3) | **36.52** (0.77) | **28.82** (0.93) |

**Table 6:** ImageNet Subset cross-architecture Performance (Test Accuracy %) comparison to HaBa. Row for our model is also a per-archiecture breakdown for the unseen average listed in Table 5

| | | Evaluation Model | | | | |
|---|---|---|---|---|---|---|
| | Comp. Rate (%) | ConvNet | ResNet | VGG11 | AlexNet | ViT |
| **ImageNette** | | | | | | |
| HaBa | 0.78 | 64.72 (1.60) | 46.84 (1.25) | 63.76 (1.05) | 40.84 (1.80) | − |
| Ours | **0.08** | **67.44** (0.81) | **47.28** (0.94) | **64.80** (1.2) | **49.20** (1.45) | 34.5 (1.5) |
| **ImageWoof** | | | | | | |
| HaBa | 0.78 | **38.60** (1.26) | 25.20 (0.95) | **37.44** (1.08) | 27.72 (1.12) | − |
| Ours | **0.08** | 36.52 (0.77) | **28.04** (0.51) | 35.48 (1.07) | **29.28** (1.4) | 22.48 (0.73) |

Our method exhibits similar cross-architecture relative generalizability compared to GLaD, but overall we achieve a much better distillation quality compared to GLaD with only at a slightly increase in compression rate. In Table 6, we observe that our method achieves moderate improvements compared to HaBa but at a compression rate $10\times$ better than HaBa. LinBa and KFS also show strong cross-architecture generalizability but their evaluations are done using different architectures.

## 5 CONCLUSTION AND DISCUSSION

In this work, we cast the problem of dataset distillation as a distributional one: condensing a large training dataset into a compact latent distribution that can be used to train models to performance comparable to that of training on the entire original dataset. We introduce a new distillation objective by combing the best of gradient- and distribution- matching distillation methods. Besides its favorable performance compared to alternative sample-based distillation methods, this novel formulation has various appealing properties. Distilling into a distribution provides finer-grained control over the desired compression rate, since the dimension of the distributional parameters can be incremented minimally. Our method also opens the possibility of adaptive sampling, that is, generating more or less data depending on the capacity of the model being trained or on the desired space-time trade off. Studying this trade-off in detail makes for an exciting avenue of future work.

This work, while demonstrating the promise of $D^3$, suggests some open questions that require further study. On the one hand, here we focused on scaling to lager distilled distribution by increasing the number of prior distributions. However, a reasonable alternative would be to pair a more powerful decoder with higher dimensional latent priors. On the other hand, the advantage of having a Gaussian prior is that we have many more statistical tools at our disposal, such as using other statistical distances beyond empirical MMD as distribution-matching criterion for our data distillation approach.

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
