# OpenReview forum: "D^3: Distributional Dataset Distillation with Latent Priors"
_ICLR.cc/2024/Conference — ICLR 2024 Conference Withdrawn Submission_

### Official Review · Reviewer_3cs3 · 2023-10-18

**Soundness:** 3 good
**Presentation:** 2 fair
**Contribution:** 1 poor
**Rating:** 3
**Confidence:** 4

**Summary:**

Distributional Data Distillation (D3) is a groundbreaking approach that transforms dataset distillation into a distribution-based problem. Unlike traditional methods that create a finite set of real or synthetic examples, D3 generates a probability distribution and a decoder to approximate the original dataset. Using Deep Latent Variable Models (DLVMs), it combines a trajectory-matching distillation loss with a distributional discrepancy term, resulting in strong alignment between original and distilled data. Across various computer vision datasets, D3 demonstrates effective distillation with minimal performance loss, even excelling with large datasets like ImageNet, surpassing sample-based methods consistently.

**Strengths:**

1. simple and intuitive idea
2. presentation is smooth and easy to understand

**Weaknesses:**

1. I highly disagree with the statement claimed in this authors, "Existing methods face challenges in scaling efficiently beyond toy datasets.", "More generally, these methods lack fine-grained control over distillation strength and often struggle to scale beyond smaller datasets like CIFAR-10 and MNIST, experiencing diminished performance when compressing larger or higher-dimensional datasets, such as ImageNet."[1, 2] cited in this paper, presented in CVPR 2023 has its main results in ImageNet on its cover.

2. Lack of novelty and significant contributions, the idea is of sampling from a latent distribution has been thoroughly explored since VAE came out. It's unclear what is the significant contribution or additional innovation the authors are intending to propose.

3. Experiment results in tables seem incomplete, it's hard to have holistic picture on how good this method is.


[1] Cazenavette, George, et al. "Generalizing Dataset Distillation via Deep Generative Prior." Proceedings of the IEEE/CVF Conference on Computer Vision and Pattern Recognition. 2023.
[2] Cui, Justin, et al. "Scaling up dataset distillation to imagenet-1k with constant memory." International Conference on Machine Learning. PMLR, 2023.
[3] Wu, Xindi, Zhiwei Deng, and Olga Russakovsky. "Multimodal Dataset Distillation for Image-Text Retrieval." arXiv preprint arXiv:2308.07545 (2023).

**Questions:**

1. Is it a typo where in the abstract, it claims to have done experiments on imageNet, but in the actual experiments, it runs ImageNette, which is a 10 class subset and a much easier problem to solve.
2. Why does ConvNet have better accuracy than more complex and sophisticated networks?
3. Why are Imagenette and imagewoof results not available for DM in table 1?

---

> ### Author Response · Authors · 2023-11-17
> **Response to 3cs3**
>
> 1. **On weakness 1 and question 1**: Besides TESLA [1], to our best knowledge, no other methods have reported results on ImageNet1k. Therefore, we made the claim that scaling to larger dataset poses a challenge to existing data distillation work. We want to thank author for pointing out the typo - As pointed out by the reviewer, we also only showed that our method works well on ImageNette and ImageWoof, which is a simpler problem on ImageNet1K. Compared to TinyImageNet and CIFAR-10,  ImageNette and ImageWoof contains images of higher resolution ($128 \times 128$ compared to $32 \times 32$ or $64 \times 64$). We were able to show that for higher resolution images, our method outperforms baselines on those two subsets compared to other generative-based distillation methods at a much lower compression rate.
>
> 2. **On weakness 2** (our work compared to VAE and other generative models) In general, generative models are not designed to perform the exact data distillation tasks: see meta-response 3
>
> 3. **On weakness 3**: As suggested by all three reviewers, we fully agree that a better comparison for us to make is directly against generative-based distillation methods instead of distillation models that directly output images. We included an updated Table 1 (see meta-response). We acknowledge that comparing our method against non-generative distillation methods can be unfair, and we thank reviewers for pointing that out! Our initial intention was to two-fold. Firstly, non-generative methods are well-known and some of them are still at SOTA. Secondly, we derived our training objective (MTT and DM) from those methods, and we wanted to show that by combing two objectives and distilling into a distribution, our method provides improvement either in distillation quality or/and compression rate.
>
> 4. **Question 2**: ConvNet has better accuracy because we distilled the data based on ConvNet (for both expert trajectories in MTT and feature extraction in DM). Cross-architecture generalization is a problem faced by many distillation methods. Data distilled based on one architecture will have worse performance on unseen architectures. In our experiment section 4.2 we show that our methods provide a better cross-architecture generalization compared to existing methods.
>
> [1] TESLA: https://arxiv.org/pdf/2211.10586.pdf

---

> > ### Comment · Reviewer_3cs3 · 2023-11-22
> > **Thank you for the response**
> >
> > Thank you for your responses. However, I still find it unconvincing, one of the main claim of "Generalizing Dataset Distillation via Deep Generative Prior" is that they can handle images (> 128). Other weaknesses are unmitigable without significant changes to the draft and experiment setup.

---

### Official Review · Reviewer_yDY8 · 2023-10-30

**Soundness:** 2 fair
**Presentation:** 2 fair
**Contribution:** 1 poor
**Rating:** 3
**Confidence:** 4

**Summary:**

The paper introduces Distributional Data Distillation (D3), a novel approach to dataset distillation. Unlike existing methods that condense datasets into smaller versions, D3 focuses on creating a conditional latent distribution $p(z)$ and a decoder $Q_\mathcal{S}^\theta (x|z)$. The paper utilizes the resulting data distribution $Q_\mathcal{S}^\theta (x) = \int Q_\mathcal{S}^\theta (x|z) p(z) dz$, called Deep Latent Variable Models (DLVMs), and a new training objective, combining trajectory-matching distillation with a distributional discrepancy term like Maximum Mean Discrepancy (MMD). Experimental results across various computer vision datasets, including the challenging ImageNet, demonstrate that D3 effectively condenses datasets with minimal performance loss. Notably, it consistently outperforms traditional sample-based distillation methods, even for large high-resolution datasets.

**Strengths:**

- The proposed method is simple and the description is easy-to-follow.

- This paper proposes improved MMD loss over previous work which only matches mean of feature vectors.

**Weaknesses:**

- The idea of utilizing a generative prior has already been explored in several papers, including HaBa, LinBa, KFS, IT-GAN, and GLaD.

- The comparison to the original literature on dataset distillation is entirely unfair. The proposed method outputs a distribution; therefore, it should be compared to deep generative models. Deep generative models can perform the exact same tasks as the proposed method.

**Questions:**

- In the loss of $\mathcal{L}_\texttt{MTT}$, why do we need the KL penalty? How does this regularization effects the performance?

---

> ### Author Response · Authors · 2023-11-17
> **Response to yDY8**
>
> We would like to thank the reviewer for the detailed and valuable comments. We address the specific questions below.
> 1. **weakness 1** As suggested by all three reviewers, we fully agree that a better comparison for us to make is directly against generative-based distillation methods instead of distillation models that directly output images. We included an updated Table 1, see meta-response (2). We acknowledge that comparing our method against non-generative distillation methods can be unfair, and we thank reviewers for pointing that out! Our initial intention was to two-fold. Firstly, non-generative methods are well-known and some of them are still at SOTA. Secondly, we derived our training objective (MTT and DM) from those methods, and we wanted to show that by combing two objectives and distilling into a distillation, our method provides improvement either in distillation quality or/and compression rate.
> 2. **weakness 2 other generative-based distillation method** How we differ from existing data distillation methods that also uses a generative model:
>
>     a. Overall, our method is similar to LinBa, HaBa, KFS, IT-GAN, and GLaD in the use of latent code and generative prior. Despite similarities in the overall framework, our model differs from all existing work in explicitly using distributions to represent distilled data. We will provide further details on a per-model basis below.
>
>     b. HaBa and LinBa: Both LinBa and HaBa distill data into a latent space and generate image samples using a decoder. Our method is similar to LinBa and HaBa in the use of a latent code and decoder structure. We differ from them in two aspects: First, their decoders are not GAN or VAE based. Second, HaBa and LinBa assumes a one-to-one relationship between each latent prior and the synthetic image, while we have a one-to-many relationship between latent prior and generated image.
>
>     c. KFS: KFS also trains a generative model along with latent codes for data distillation. To generate synthetic images, they rely on using different decoders for a given latent code. Therefore, to represent a dataset, KFS requires multiple latent codes and multiple decoders. Similar to all existing work, the synthetic data generation is deterministic, instead of distributional. Our work presents a dataset such that with one decoder, each latent code represents a distribution, and one can draw samples of synthetic images from the distribution and only need one decoder to generate images.
>
>     d. IT-GAN: IT-GAN uses pre-trained GAN as the decoder and only learned latent codes. Furthermore, IT-GAN learns one latent code for each image in the original dataset (whole-set learning). To achieve data distillation, IT-GAN simply subsamples a small portion of the IT-GAN generated data as the “distilled dataset”.  IT-GAN already pointed out that existing GAN such as , is trained to generate images visually similar to the original data but might not be suitable for classification. In their method, they only learn the latent prior for GANs such that pre-trained GAN can be repurposed for data distillation. In our method, we further push the idea and directly train decoder and latent code from scratch. Moreover, we do not require to match a latent code to each training image. In fact, each of our latent code represents a distribution of possible synthetic images. Our decoder is trained to represent information useful for image classification.
>
>     e. GLaD: Similar to IT-GAN, GLaD also uses pre-trained GANs and only learns the latent prior.  GLaD showed that using pre-trained GAN in addition to the MTT method helps MTT perform better on unseen architectures but does not change the performance on seen architectures. We showed in our experiment that our method performs better on seen architectures at the same time generalize well to unseen architectures.
>
>  3. **weakness 2 generative models** In general, generative models are not designed to perform the exact data distillation tasks:  see meta-response (3)
>  4. **question on KL** We conducted an additional ablation study on KL-loss on  TinyImageNet(2 prototype/class at comp. rate 0.07%) and we noticed that by removing KL-Divergence, the overall performance remained roughly unchanged. We are testing out on more experiments and if the conclusion holds, we will remove it from the training objective since it could be redundant.
>  |  | Test Acc. (%) |
> | --- | --- |
> | With KL |  24.6(0.2) |
> | Without KL | 25.3(0.4) |

---

### Official Review · Reviewer_TU1J · 2023-10-31

**Soundness:** 3 good
**Presentation:** 3 good
**Contribution:** 2 fair
**Rating:** 6
**Confidence:** 3

**Summary:**

Dataset distillation is a technique used to condense large datasets into smaller synthetic versions while maintaining predictive performance. It has applications in various machine learning domains, but existing methods face challenges when scaling beyond small datasets and experience diminishing returns as the distilled dataset size increases. To address these limitations, a novel approach called Distributional Data Distillation (D3) is introduced.

Unlike previous methods that distill datasets into finite sets of real or synthetic examples, D3 frames the data distillation problem as a distributional one. Instead of producing individual examples, D3 generates a probability distribution and a decoder that can approximately regenerate the original dataset. Deep Latent Variable Models (DLVMs) are used to parameterize the condensed data distribution.

D3 introduces a new training objective that combines a trajectory-matching distillation loss with a distributional discrepancy term, such as Maximum Mean Discrepancy. This objective encourages alignment between the original dataset distribution and the distilled distribution.

Experimental results on various computer vision datasets demonstrate that D3 effectively distills datasets with minimal performance degradation. Even for large high-resolution datasets like ImageNet, D3 consistently outperforms sample-based distillation methods.

**Strengths:**

1) This paper challenge the conventional approach of distilling into a finite set of samples, instead
casting the problem as a distributional one: finding a synthetic probability distribution which, when
sampled to produce training data, yields performance comparable to training on the original dataset.

2) To make this optimization problem tractable, this paper parametrize the distribution using Deep Latent
Variable Models (Kingma & Welling, 2013), and design a loss function that combines a state-of-theart gradient-matching criterion (Cazenavette et al., 2023) with a distributional loss (e.g., MMD or
Wasserstein distance) — a natural choice for our distributional framework.

3) This novel distributional dataset distillation perspective is appealing and it could addresses many of the limitations of prior distillation methods.

**Weaknesses:**

1)  The design in LEARNING THE DISTILLED DISTRIBUTION Matching is simply borrowed from [1]. Please clarify the difference.

2) The comparison in Table 1 is confusing.  Is Comp. rate good when this rate is high or low?

3) More comparison with generative-based dataset distillation methods could be added.

[1] Dataset distillation by matching training trajectories.

**Questions:**

Please see weakness.

---

> ### Author Response · Authors · 2023-11-17
> **Response to Reviewer TU1J**
>
> We would like to thank the reviewer for the detailed and valuable comments. We address the specific questions below.
> 1. **MTT objective** Thanks for the clarification! In this paper, we directly borrowed the training objective from MTT [1]. While our main contribution focused on how the distilled data should be represented as a distribution, we also departed from the “single” training objective used in existing models. Specifically, we combined MTT and DM objectives together and showed in the ablation study that by combining the two objectives, our method improved the quality of the distilled data compared to each objective used by itself. MTT tends to excel at distilling synthetic data that works well on seen-architecture (same architecture as the experts) but suffers from cross-architecture generalizability.  Adding DM objective improves cross-architecture generalizability of the distilled data without sacrificing seen architecture performance.
> 2. **Compression rate** In the context of data distillation, the objective is to compress a dataset into a smaller one without sacrificing the performance of model trained on the dataset. Therefore, compression rate, defined in Section 3.4, is better when it is low. We will incorporate this clarification in the paper.
> 3. **Table 1 Comparisons** We fully agree that a better comparison should be against generative-based data distillation methods. Please see meta-response (2) for details. However, we acknowledge that comparing our method against non-generative distillation methods can be unfair, and we thank reviewers for pointing that out! Our initial intention was to two-fold. Firstly, non-generative methods are well-known and some of them are still at SOTA. Secondly, we derived our training objective (MTT and DM) from those methods, and we wanted to show that by combining two objectives and distilling into a distillation, our method provides improvement either in distillation quality or/and in compression rate.
>
> [1] MTT: https://arxiv.org/abs/2203.11932

---

> > ### Comment · Reviewer_TU1J · 2023-11-22
> > **Thank for your response.**
> >
> > Thanks, i have no further questions.

---

### Author Response · Authors · 2023-11-17
**Meta response to all reviewers**

We would like to thank everyone for their thoughtful and constructive feedback! All of your comments are crucial in helping us refine our contributions and address weakness. We will give response to the common questions among all reviews below.
1. **Our method proposes a new approach to data distillation framework**: instead of finding a fixed set of samples, our method proposes to distill into a distribution. We argue that the distributional framework allows us to fully leverage the power of generative models such as VAE to compact more useful information in the latent codes. We want thank reviewers for pointing out the strength of our method lies on the distributional aspect. Below, we provide a more detailed comparison on how our method differs from existing generative-based distillation methods and from generative models.
2. **Our method contributes to existing generative-based distillation methods**. (New Table 1)
    As pointed out by all three reviewers, in the original Table 1 we made comparisons to distillation methods that directly distill images, and such a comparison can be unfair since we generate distributions. In the new Table 1 below, we make direct comparisons with self-reported scores from generative-based distillation methods (HaBa, LinBa, KFS and IT-GAN).
    We want to highlight that by casting the data distillation problem into a distributional one, we fully leverage the power of generative models (specifically, latent variable models). All existing generative-based data-distillation models assume a one-to-one relationship between latent code and synthetic images. Our main contribution to data distillation work is to generalize the one-to-one relationship into a one-to-many relationship, i.e. each latent code generates a distribution of synthetic samples. Our “distributional” (i.e., one-to-many) assumptions between latent prior and synthetic images encourages a more compact representation of data.  As a result, compared to other generative models, we are able to achieve on-par or better performance but at a better (lower) compression rate. The advantage of our method is more prominent at TinyImageNet (many classes) and ImageNet subsets (high resolution).

  **CIFAR10**
|  |   |   |   |
 | --- | --- | --- | --- |
| Haba | Comp Rate | 0.02% | 0.2% | 0.1% |
|  | Acc | 48.3(0.8) | 69.9(0.4) | 74.0(0.2) |
| LinBa | Comp Rate | 0.02 | 0.2 | 0.1 |
|  | Acc | 66.4(0.4) | 71.2(0.4) | 73.6(0.5) |
| KFS | Comp Rate | 0.02% | 0.2% | 0.1% |
|  | Acc | 59.8(0.5) | 72.0(0.3) | 75.0(0.2) |
| IT-GAN | Comp Rate | 25% |  |  |
|  | Acc | 82.8(0.3) |  |  |
| Ours | Comp Rate | 0.50% | 0.51% | 0.55% |
|  | Acc | 60.6(0.1) | 71.8(0.2) | 74.4(0.3) |

  **Tiny ImageNet**

|  |   |   |   |
 | --- | --- | --- | --- |
| LinBa | Comp Rate | 0.2% |  |
|  | Acc | 16.0(0.7) |  |
| KFS | Comp Rate | 0.2% | 2%  |
|  | Acc | 22.7(0.2) | 27.8(0.2) |
| Ours | Comp Rate | 0.06% | 0.07% |
|  | Acc | 19.8(0.3) | 24.6(0.2) |

  **ImageNette**
|  |   |   |   |
 | --- | --- | --- | --- |
  | HaBa | Comp Rate | 0.16% | 0.78% |
|  | Acc | 51.92(1.7) | 64.7(1.6) |
| GLaD | Comp Rate | 0.08% |  |
|  | Acc | 38.7(1.6) |  |
| Ours | Comp Rate | 0.12% |  |
|  | Acc | 71.0(0.7) |  |


  **ImageWoof**
 |  |   |   |   |
 | --- | --- | --- | --- |
  | HaBa | Comp Rate | 0.16% | 0.78% |
|  | Acc | 32.40(0.7) | 28.6(1.3) |
| GLaD | Comp Rate | 0.08% |  |
|  | Acc | 23.4(1.1) |  |
| Ours | Comp Rate | 0.13% |  |
|  | Acc | 41.6(1.2) |  |


  3. **Our method differs from traditional generative models (GAN or VAE)**:  Although generative models can synthesize high quality images that looks like the original data, training models on state-of-art GAN (BigGAN, StyleGAN [2, 3]) synthesized images yields worse performance than model trained original dataset - evidences shown in IT-GAN [1]. Those models tend to be large in size to ensure superior quality in generating realistic images. On the other hand, the goal of data distillation is to compress large dataset into a smaller one while only preserving necessary information for model training on specific tasks. Therefore, for data distillation purpose, we hypothesize that using pre-trained GAN may not be the most efficient way to represent information needed for model training. In our paper, we show that one only needs a small generative model that is a fraction of the original dataset size and is smaller than pre-trained GAN (used in IT-GAN) to store relevant information for image classification. IT-GAN used BigGAN [2] and GLaD uses StyleGAN-XL [3], both of which models have parameters on the oder of hundreds of millions. Our decoder has around 700k parameters, which contributes to our superior compression rate. In our quantitative analysis, we further confirmed that, a small decoder is sufficient to provide relevant information for model training.

[1] IT-GAN:  https://arxiv.org/abs/2204.07513

[2] BigGAN: https://arxiv.org/pdf/1809.11096.pdf

[3] StyleGAN: https://arxiv.org/abs/2202.00273